# Optical emissions associated with narrow bipolar events from thunderstorm clouds penetrating into the stratosphere

Feifan Liu[1], Gaopeng Lu[1], Torsten Neubert [2], Jiuhou Lei [1,3✉], Oliver Chanrion[2], Nikolai Østgaard[4], Dongshuai Li [5], Alejandro Luque [5], Francisco J. Gordillo-Vázquez[5], Victor Reglero[6], Weitao Lyu[7] & Baoyou Zhu[1✉]

Narrow bipolar events (NBEs) are signatures in radio signals from thunderstorms observed by ground-based receivers. NBEs may occur at the onset of lightning, but the discharge process is not well understood. Here, we present spectral measurements by the Atmosphere-Space Interactions Monitor (ASIM) on the International Space Station that are associated with nine negative and three positive NBEs observed by a ground-based array of receivers. We found that both polarities NBEs are associated with emissions at 337 nm with weak or no detectable emissions at 777.4 nm, suggesting that NBEs are associated with streamer breakdown. The rise times of the emissions for negative NBEs are about 10 μs, consistent with source locations at cloud tops where photons undergo little scattering by cloud particles, and for positive NBEs are ~1 ms, consistent with locations deeper in the clouds. For negative NBEs, the emission strength is almost linearly correlated with the peak current of the associated NBEs. Our findings suggest that ground-based observations of radio signals provide a new means to measure the occurrences and strength of cloud-top discharges near the tropopause.

[1] CAS Key Laboratory of Geospace Environment, School of Earth and Space Sciences, University of Science and Technology of China, Hefei, China. [2] National Space Institute, Technical University of Denmark (DTU Space), Kongens Lyngby, Denmark. [3] CAS Center for Excellence in Comparative Planetology, University of Science and Technology of China, Hefei, China. [4] Birkeland Centre for Space Science, Department of Physics and Technology, University of Bergen, Bergen, Norway. [5] Instituto de Astrofísica de Andalucía, CSIC, Glorieta de la Astronomia s/n, Granada, Spain. [6] Image Processing Laboratory, University of Valencia, Valencia, Spain. [7] State Key Laboratory of Severe Weather, Chinese Academy of Meteorological Sciences, Beijing, China. ✉email: leijh@ustc.edu.cn; zhuby@ustc.edu.cn

N arrow bipolar events (NBEs), also known as compact intracloud discharges (CIDs), are one special class of intra-cloud (IC) fast discharges that have been considered as one of the most intriguing phenomena receiving enormous interest from many lightning researchers[1–5]. They are characterized with short-duration (typically 10–30 μs) electromagnetic waveforms and strong radiation in the high-frequency (HF) and very high-frequency (VHF) bands[6,7]. NBEs often appear isolated from other IC discharges[1,8], but in many cases they also occur as the initial event of ordinary bi-level IC lightning flashes[4,9,10]. The experimental and modeling work suggests that they are formed by streamers inside thunderclouds[11–14].

In the normally electrified thunderstorms, NBEs of positive and negative polarities are distinctly segregated into thundercloud regions centered at different altitudes[2,3,6,15]. The positive NBEs

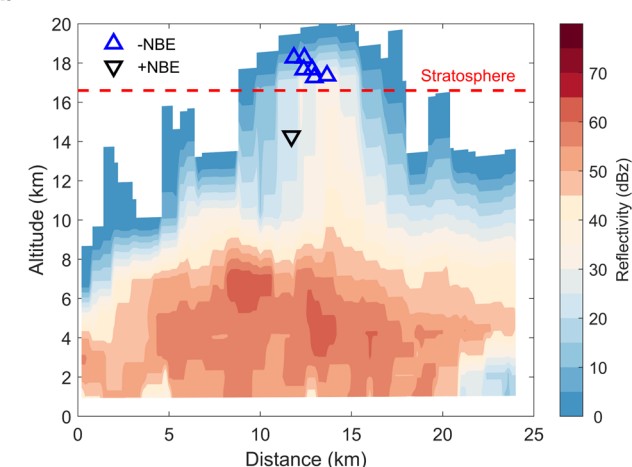

**Fig. 1 Meteorological condition of the parent thunderstorm producing both polarities NBEs observed on 7 August 2019. a** Cloud top temperatures and lightning locations of the thunderstorm. Color shade represents temperature variations of cloud top. The black line presents the ISS orbit and the red dot represents the ASIM at 13:06:000 UTC. The green '★' represents the GZ lightning VLF/LF receiver (113.615°E, 23.568°N). The pink '×' denotes locations of NBEs. The gray point represents lightning locations obtained from the Vaisala LS7000 lightning location networks. The cyan '+' represents the Shenzhen S-band Doppler radar station (114.005°E, 22.542°N). **b** The vertical profile of the radar reflectivity along the dotted cyan line at 13:06 UTC. Color shade presents reflectivity variations. The red horizontal line represents the local tropopause height at about 16.6 km obtained from the sounding data. Source data plotted for this figure are provided in the Source data file.

usually occur between the main negative and the upper positive charge layer of thunderclouds[2,9,16]. The negative NBEs are far from being as frequent as their positive counterpart, whereas they are usually produced at much higher altitude between the upper positive and the negative screening layer with thundercloud tops typically reaching over 16 km[17,18], and thus serve as a proxy to monitor the global deep convection. Additionally, they have been thought to trigger the blue emissions[18,19], one kind of electrical discharges shooting upward from the thundercloud top into the stratosphere[20–25], which can perturb the concentrations of greenhouse gases near the tropopause[26–28].

For decades the luminous features of NBEs were a mystery and somewhat under debate. NBEs have been thought to be relatively "dark" (non-luminous) compared to other fast lightning discharges[29–31]. Jacobson et al.[32] compared 193 positive and 24 negative NBEs from the Los Alamos Sferic Array (LASA) with the optical observations from FORTE. It was found that NBEs of both polarities had dim optical emissions in the range of 400–1100 nm (roughly from violet to far infrared). Recently, the Modular Multi-spectral Imaging Array (MMIA) of the Atmosphere-Space Interaction Monitor (ASIM) aboard the International Space Station (ISS) provided the multi-spectral observations of thunderstorm discharges with the highest spatial (~400 m/pixel) and temporal (10 μs) resolution so far[33]. Positive NBEs inside thunderclouds were found to be associated with the 337 nm emissions rather than the 777.4 nm emissions, suggesting that they are corona-like discharges formed by numerous cold streamers[34]. However, this leads to a question of what is the optical signature of negative NBEs occurring near the thundercloud top[25]. The optical features of NBEs still merit verification.

Here we present the observations of NBEs with both polarities by a ground-based lightning detection array at the closest distance of around 100 km and the associated optical blue emissions by ASIM with an unprecedented time resolution of 10 μs. Our analyses show the distinct optical signals of negative NBEs and the relationship between the blue emissions and radio signal of negative NBEs. This indicates the direct link of blue emissions to NBEs and confirms the suspicion of Neubert et al.[25] that the microsecond blue emissions could be the optical equivalent of negative NBEs.

## Results

**Meteorological condition of the parent thunderstorm.** On the evening of 7 August 2019, during a short time period between 13:05:56 and 13:06:32 UTC, MMIA recorded an outbreak of 12 blue emissions over a compact thunderstorm near the coastline of South China. We compared the trigger time of blue emissions with the data recorded by a ground-based very-low-frequency/low-frequency (VLF/LF) network of multiple stations (detailed time-match information is provided in "Methods" and Table S1). It is found that nine of these 12 blue emissions are associated with negative NBEs, and the other three are with positive NBEs. To the best of our knowledge, it is the first ever report of NBEs with both polarities produced in one thunderstorm observed from space and ground-based radio sensors at a closest distance of about 100 km. Among these events, the height of six negative NBEs and one positive NBE can be determined through the ionospheric reflection pair[7,35]. The source altitude of negative NBEs was located at about 18 km (above mean sea level, MSL) near the cloud top, while the positive NBEs occurred at an altitude of about 14 km inside the thundercloud, which is consistent with previous studies on the height distribution of NBEs[9,15,17], indicating that the higher negative NBEs are inferred to initiate between upper positive charge region and screening negative charge layer[15,17].

Figure 1a shows the lightning activity of the thunderstorm overlapped on the infrared (IR) brightness temperatures at

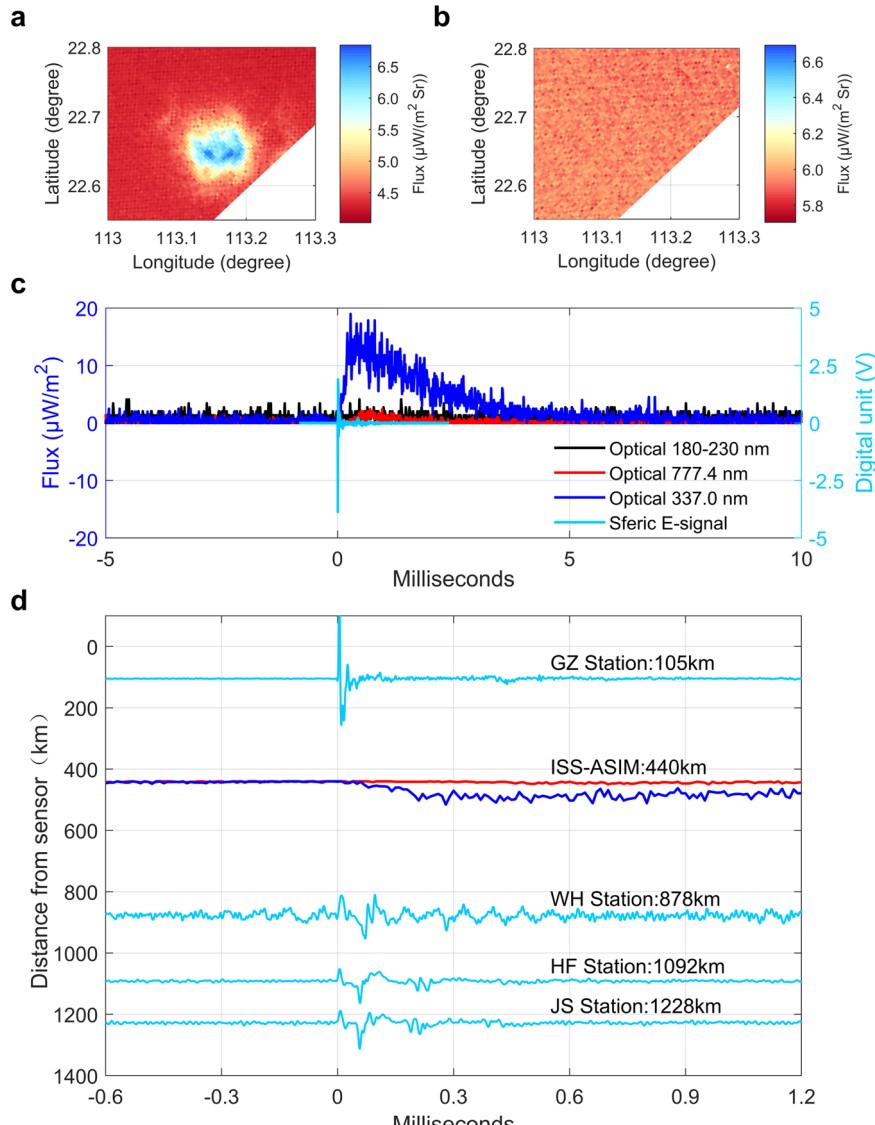

**Fig. 2 Optical and electrical signature of a positive NBE. a** ASIM camera image in the 337 nm filter; **b** Camera image in the 777.4 nm filter. Color shade represents flux variations on logarithmic scale. **c** Comparison between the blue emissions detected by MMIA and the VLF/LF sferic waveforms recorded at GZ station. $t = 0$ ms corresponds to 13:06:02.691042 UTC on 7 August 2019. **d** Details of the optical signal and VLF/LF sferic waveform at four stations. The VLF/LF sferic signal is shown as the cyan line. The 337 nm and 777.4 nm emissions are plotted with blue and red lines, respectively. Source data plotted for this figure are provided in the Source data file.

10.4 μm derived from the geostationary meteorological satellite Himawari-8 at 13:10 UTC. NBEs occurred near the region of the coldest cloud top with a temperature of 190 K. As shown in Fig. S1, the inferred tropopause height from the sounding data is at about 16.6 km. Figure 1b shows the vertical profile of radar echo of the thunderstorm at 13:06 UTC. The highest altitude of reflectivity reaches nearly 18 km, showing the penetration of thundercloud into the stratosphere. The evolution of the radar reflectivity through the parent thunderstorm is shown in Fig. S2. It can be seen that NBEs were produced under the intense convective surges. The negative NBEs occurred in the region with reflectivity 15 dBZ at the cloud top, while the positive NBEs were located relatively deeper in the thundercloud.

**Optical and electrical observations of positive NBEs**. Figure 2 presents the optical blue emissions recorded by MMIA at 13:06:02.691042 UTC and the associated positive NBE waveform detected by four VLF/LF stations at range of 105 km to 1228 km.

The height of this NBE is determined to be about 15 km (MSL) by calculating the time difference between the ground wave and two ionospheric reflections. It can be seen that the positive NBE is associated with the blue emissions centered at 337 nm, which is consistent with Soler et al.[34] who reported that seven positive NBEs located between 8 and 15 km inside thunderclouds are associated with the 337 nm emissions. This suggests that positive NBEs are corona discharges formed by numerous streamers, confirming previous work by Tilles et al.[11] and Liu et al.[12].

Additionally, this positive NBE also gave off weak 777.4 nm emission. The context of these events plotted in Fig. 2c shows that the main pulse of NBE signals is followed by some extra sferic-radiating activity, indicating that the NBE was not the only discharging activity. Thus, it is likely that the 777.4 nm emission is equally related to the additional discharging activity but not related to the NBE. Jacobson et al.[32] examined the optical emission of 193 positive and 24 negative NBEs based on the measurements of FORTE and LASA. It was found that no obvious optical emissions were associated with NBEs except for two

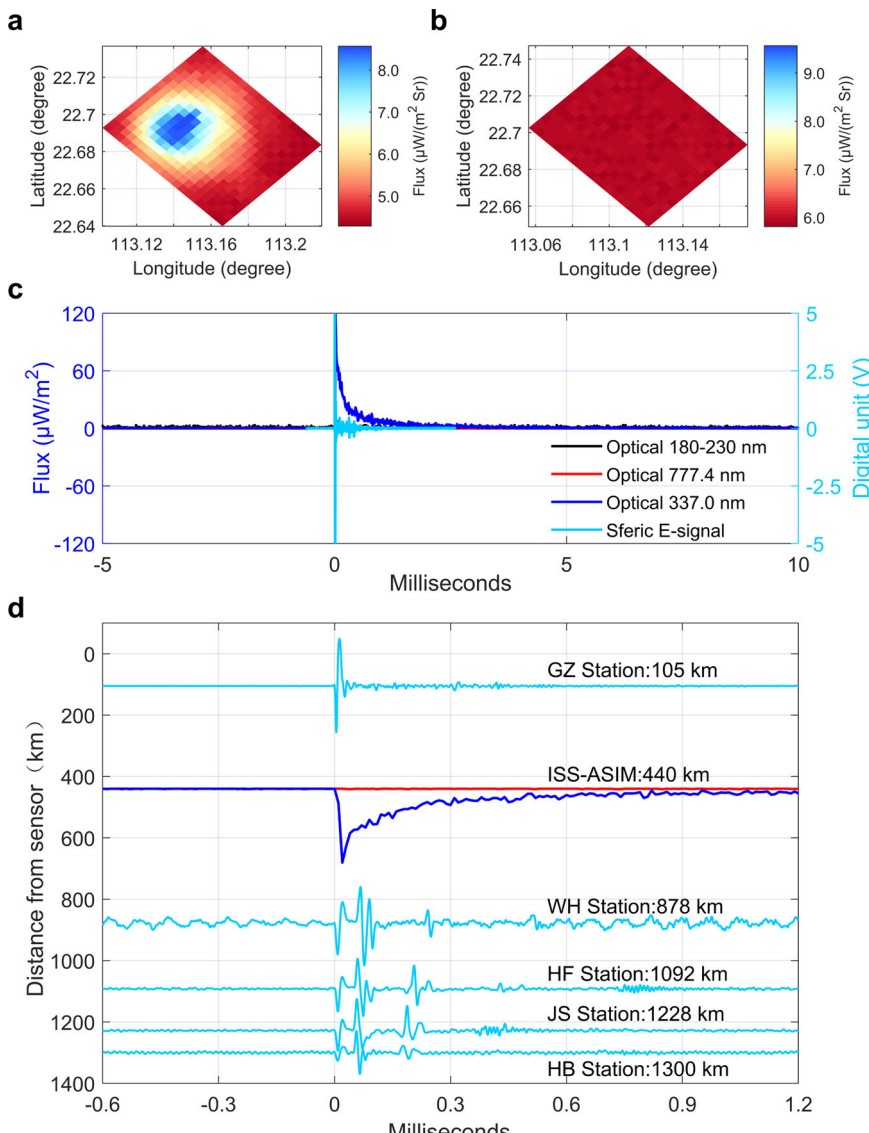

**Fig. 3 Optical and electrical signature of a negative NBE. a** ASIM camera image in the 337 nm filter; **b** Camera image in the 777.4 nm filter. Color shade represents flux variations on logarithmic scale. **c** Comparison between the blue emissions detected by MMIA and the VLF/LF sferic waveforms recorded at GZ station. $t = 0$ ms corresponds to 13:06:20.968870 UTC on 7 August 2019. **d** Details of the optical signal and VLF/LF sferic waveform in five stations. The VLF/LF sferic signal is shown as the cyan line. The 337 nm and 777.4 nm emissions are plotted with the blue and red lines, respectively. Source data plotted for this figure are provided in the Source data file.

positive NBEs. They examined the context of these two NBE events and found that their main sferic pulses were both preceded by some earlier sferic activity. Therefore, the 777.4 nm optical emission is likely related to additional sferic activity accompanying the NBE itself. The remaining two positive NBEs shown in Fig. S4 generally exhibit similar optical features that include a distinct 337 nm emission and no 777.4 nm signal above the noise level. The distant stations failed to record the signal of two positive NBEs due to the relatively small amplitude, and their height also cannot be inferred from the radio signal recorded at the Guangzhou (GZ) station due to the weak ionospheric reflection.

**Optical and electrical observations of negative NBEs.** Figure 3 compares the blue emissions recorded by MMIA at 13:06:20.968870 UTC and the associated negative NBE waveform measured at five VLF/LF stations located at a range of 105 km to 1300 km. The waveforms recorded at four relatively distant stations (>800 km)

exhibit the obvious ionospoheric reflection that can be used to determine a source height of about 18 km (MSL). It can be seen that this negative NBE was also associated with the 337 nm emission, but there was no significant 777.4 nm signal. The emissions at 777.4 nm are from atomic oxygen and are one of the major lines of the lightning leader spectrum, suggesting that negative NBEs are associated with streamer breakdown, similar to the positive NBE. An evaluation, based on optical radiation transfer, of the streamers of these NBEs is reported in a complementary publication, which presents the streamer-like structure of negative NBEs that typically involves around $10^9$ streamer initiation events[36]. Here it is emphasized that the negative NBE produced the distinct 337 nm optical signature with rise time of <0.05 ms, showing the much sharper optical emission of this negative NBE than positive NBEs. In addition, we can see that the VLF/LF signal of negative NBEs corresponds to the rise stage of 337 nm emissions. For the remaining eight negative NBEs shown in Fig. S4, our analyses generally obtain the similar optical feature, namely the rise time of

**Table 1 Detailed information of the optical and electrical signature of the NBE observations.**

| Optical trigger time (UTC) | Optical rise (ms) | Optical duration (ms) | Peak brightness (µW/m²) | Optical SNR | Type | VLF_Lat | VLF_Lon | VLF Height (km) | Peak Current (kA) | VLF Duration (ms) |
|---|---|---|---|---|---|---|---|---|---|---|
| 13:05:56.937910 | 0.05 | 4.64 | 20.11 | 22.4 | −NBE | 22.662 | 113.226 | | −11.32 | 0.06 |
| 13:05:58.633000 | 0.03 | 3.80 | 142.18 | 35.8 | −NBE | 22.686 | 113.266 | 18.0 | −118.82 | 0.04 |
| 13:06:01.757770 | 0.07 | 3.73 | 30.03 | 32.6 | −NBE | 22.675 | 113.224 | | −13.47 | 0.08 |
| 13:06:02.691042 | 0.31 | 6.08 | 16.40 | 21.9 | +NBE | 22.671 | 113.275 | 14.6 | +82.30 | 0.06 |
| 13:06:09.569330 | 0.03 | 3.20 | 97.88 | 33.5 | −NBE | 22.687 | 113.279 | 18.0 | −117.71 | 0.05 |
| 13:06:16.634460 | 0.03 | 5.35 | 44.51 | 29.6 | −NBE | 22.677 | 113.253 | 17.6 | −29.58 | 0.05 |
| 13:06:17.579338 | 0.03 | 3.62 | 11.50 | 21.8 | −NBE | 22.677 | 113.253 | | −10.14 | 0.04 |
| 13:06:20.968870 | 0.03 | 3.31 | 120.30 | 41.2 | −NBE | 22.668 | 113.224 | 17.7 | −135.30 | 0.03 |
| 13:06:23.435358 | 0.20 | 6.13 | 5.60 | 17.3 | +NBE | 22.669 | 113.272 | | +8.65 | 0.05 |
| 13:06:26.164829 | 0.34 | 6.43 | 7.20 | 20.2 | +NBE | 22.824 | 112.994 | | +12.74 | 0.05 |
| 13:06:30.494520 | 0.03 | 4.75 | 46.48 | 29.2 | −NBE | 22.676 | 113.234 | 18.6 | −65.30 | 0.04 |
| 13:06:31.657420 | 0.03 | 3.86 | 39.96 | 26.5 | −NBE | 22.682 | 113.224 | 18.6 | −69.68 | 0.03 |

the 337 nm emissions for the negative NBEs is much shorter than that of positive NBEs.

To characterize the waveform parameters of the optical signals, we define the rise time, duration, and signal-to-noise ratio (SNR) (definition of rise time, duration, and SNR is given in Fig. S3 of the Supplementary information). Table 1 summarized the detailed parameters of optical signature and VLF/LF sferics for these NBEs. The observed negative NBEs have a wide range of amplitude. Previous observations show that the fast breakdown of NBEs occurs with an extremely wide range of strengh, both in VHF and VLF/LF bands, while still initiating ordinary IC discharges[4,11]. We can see that the optical rise time of negative NBEs is in the range of 0.03 ms-0.08 ms, which is much shorter than the positive counterparts (>0.2 ms in all three cases). For the whole optical duration, there is also a considerable difference that in all cases positive NBEs endure longer than the negative ones. The reason why negative NBEs appear much sharper than positive NBEs in the optical emission is discussed below.

**Comparison between blue emission and radio signal of NBEs.** As the negative NBEs in our obervations occur close to the cloud top, its optical radiation is less affected by the thundercloud scattering[37,38]. This allows us to compare by means of radio observations. As shown in Fig. 3, we see that the VLF/LF signal of negative NBEs corresponds to the onset of 337 nm blue emission, but the radio signal of NBEs in the VLF/LF band is much shorter than their optical blue emission. The radio signal of NBEs is usually shorter than 0.03 ms, while the optical duration of the associated blue emissions is usually longer than 3 ms. This suggests that the lifetime of current source is much shorter than the optical pulses.

To characterize the source current further from optical measurements and compare the VLF/LF signal of negative NBEs with the 337 nm emission in association, we implement the first- and second-order time-derivative on the original 337 nm optical signal shown in Fig. 4a. Since the sampling rate of the optical signal is 100 kHz, its maximum bandwidth is 50 kHz. To eliminate the influence of bandwidth, we apply a 50-kHz lowpass filter to the VLF/LF signal. It is seen that the second-order derivative of 337 nm emission is very similar to the 50-kHz lowpass filtered waveform of negative NBEs. The second-order derivative of 337 nm emission can be divided into two parts. The first part is the main narrow bipolar waveform, which resembles the sferic waveform of negative NBEs in the VLF/LF band; the second part is the one in which small oscillation pulses ensue, and these small pulses are also discernible on the VLF/LF signal owing to the close distance (105 km) of the GZ station (the VLF/LF waveform for other cases is given in Figs. S5–S12). These small pulses after the main bipolar pulse of the VLF/LF signal suggest that there still exists some weak current pulses after the fast breakdown[4], which corresponds to the slow descent stage of 337 nm blue emissions. The electric field (E-field) changes observation by Karunarathne et al.[39] suggest that NBE has static offset in its bipolar pulse and was also followed by a slow electrostatic change lasting about several ms, which is similar to our optical observation. Therefore, the duration of NBEs could be as long as several ms as observed by the optical detector, but on the VLF/LF waveform, the subsequent small oscillation pulses may be attenuated due to distance.

The question is why the second-order derivative of the optical signal exhibits a similarity with the lowpass-filtered VLF/LF waveform. The luminosity is roughly proportional to the electrical current intensity $(I)$[40]. The VLF/LF signal at the GZ station is the time derivative of the vertical E-field $(dE_z/dt)$. The measurement distance of about 105 km for the VLF/LF signal might be very critical for finding this waveform similarity. As the

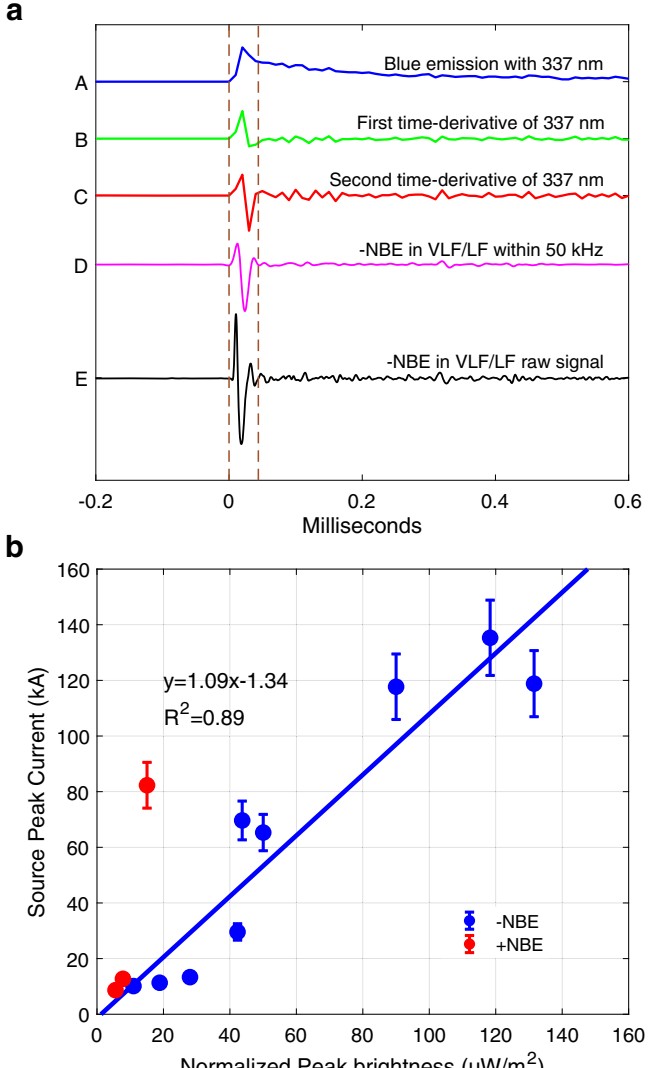

**Fig. 4 Comparison for the VLF/LF signal of a negative NBE and optical blue emissions in 337 nm. a** Comparison between raw (E) and 50 kHz filtered (D) VLF/LF signal of negative NBE, and blue emissions (A), and their first (B) and second (C) time-derivative. **b** Correlations between the peak brightness of blue emission and peak current of NBEs. Note that each peak brightness of 337 nm was normalized to 450 km. Vertical bars present the err bars of the estimated peak current for NBEs (see "Methods"). Source data plotted for this figure are provided in the Source data file.

*E*-field is also proportional to the temporal derivative of *I* at this distance[41,42], it is reasonable that the second-order derivative of the optical signal is similar to the lowpass-filtered VLF/LF signal of negative NBEs.

## Discussions

Soler et al.[34] reported seven positive NBEs located at heights of 8–15 km (MSL) in the thundercloud that were all associated with the 337 nm blue emissions, and there was no 777.4 nm emission based on the multi-spectral data of MMIA. They suggested that blue flashes are from cold ionization waves (associated with streamers) and the positive NBEs are corona discharges formed by many streamers. For our observations, both positive and negative NBEs produced by the same thunderstorm were observed by ASIM and the ground-based sferic array. It is found that negative NBEs are also associated with 337 nm, without 777.4 nm, either, suggesting that both polarities of NBEs are

streamer discharges, which confirms previous results that fast breakdown is formed by streamers, as inferred by Liu et al.[12].

The optical rise time of positive and negative NBEs exhibits a significant difference; that is, the optical rise time of negative NBEs appears to be much shorter than that of positive NBEs. It is speculated that there might be two reasons causing this distinct difference: the main reason is likely that photons are scattered in cloud particles, which tend to broaden pulses and increase their rise times. This effect becomes more pronounced the deeper in the cloud the source is located[34,37,38]. Light et al.[37] simulated the optical waveform of lightning for the satellite observations and discussed the effects caused by the optical depth, the position of light emission, and the size and shape of thundercloud. They set the source position at varying heights below the top of the thundercloud, and found that the rise time at the higher altitude is generally sharper than that of the event inside the thunder-cloud. Light et al.[43] also showed that the peak amplitude of lightning event could be attenuated by up to 2–3 orders of magnitude. Negative NBEs, when they occur in the upper levels of clouds, are usually at the very top, and the optical emissions observed from space are therefore little affected by scattering. Positive NBEs, on the other hand, typically occur at some depth in clouds, and their emissions, when observed from space, have longer rise times and are damped by absorption.

A second effect on the rise time may relate to the timescales of the sources. If a discharge sources is at the cloud top, the observed rise time is a measure of the source rise time. The rise times for negative NBEs are 10–30 μs, suggesting that the full discharge length is established during this time. If the discharge propagates from one segment of the cloud to another with a typical streamer speed from $1 \times 10^5$ to $2 \times 10^7$ m s$^{-1}$ [44,45], its channel length is likely no longer than a few hundred meters. The source onset and discharge length of positive NBEs deeper in the cloud have as yet not been estimated from optical pulses because of the dominance of cloud scattering, but the duration of an NBE in our radio observation indicates that it seems to be independent of its polarity. As NBEs play a role in the initiation of lightning, it is of interest to study more observations of NBEs and blue flashes, to arrive at a better characterisation of their properties.

Space-borne optical observations provide an effective tool for characterizing the global lightning activity. However, due to the lack of comparison with the ground-based sferic data, it is difficult to determine the specific lightning type from the optical signal. Our findings that the distinct optical signals of NBEs with different polarities would provide a new tool to identify and distinguish NBEs of both polarities based on a space-borne optical observation platform and further provide the possibility to evaluate the global NBE occurrence.

Furthermore, we compare the peak optical brightness of 337 nm and peak current estimated from the radio signal for both positive and negative NBEs. To eliminate the influence of distance, each peak of 337 nm emission was normalized to 450 km. Figure 4b shows the relationship between the peak brightness and peak current of NBEs. Although the number of samples is relatively limited, there exists a clear correlation between the peak 337.4 nm emission and peak current of the associated NBEs, except for one positive NBE with a high peak current (about +80 kA). The deviation is likely caused by the attenuation of thunderclouds. This correlation is also observed in "normal" lightning[46,47]. Kikuchi et al.[47] examined the relationship between the photometer (PH4) at wavelengths of 599–900 nm and electromagnetic waveforms in the LF band for 11 negative cloud-to-ground (CG) strokes. They also found a linear correlation between the absolute optical intensity and the peak current.

Negative NBEs have been thought to be the initial event of blue emissions due to the relatively low time resolution of previous

optical observations[18,19]. The optical waveform obtained with an unprecedented 10-$\mu$s resolution and the peak correlation analyzed above present direct evidence that blue emissions are the optical signature produced by negative NBEs. Blue emissions usually occur in the strong thunderstorms with overshooting tops penetrating into the lower stratosphere[20,21,24,25,48]. Such discharges (that can be precursors of blue jets) can affect the exchange of greenhouse gases between the troposphere and stratosphere through the production of nitrous oxide, and the depletion of stratospheric ozone[28,49,50,51]. Winkler and Notholt[50] estimated the production of nitrous oxides to be about $10^{12}$ cm$^{-3}$ of a blue jet initiating at 18 km. The production of the kilometer-scale blue discharges would lead to 1660 mole per discharge[24]. However, this region of atmosphere is difficult to access experimentally. Our results present the direct link between the blue emissions and radio signal of negative NBEs. On that perspective, ground-based observations of radio signals would provide a possible new solution to measure the occurrences and strength of cloud-top discharges with implications for studying the perturbations of greenhouse gas concentrations near the tropopause.

## Methods

**Optical measurements.** The optical emission data analyzed in this study are obtained from the MMIA instrument of ASIM aboard the International Space Station (ISS) at an altitude of about 400 km since 2 April 2018. So far it is the space-borne platform at the lowest altitude to study lightning activity in tropospheric thunderstorms and their effects in the near-Earth space[33]. MMIA consists of two parts, including a pair of filtered cameras (operating at 12 frames per second) and three photometers[52]. The filtered cameras offer the spatial resolution of optical events, one in the near UV at the spectral line of the $N_2$ second positive system ($N_2$ SPS, 337 nm/4 nm) and the other in one of the strongest lightning emissions corresponding to neutral oxygen (777.4 nm/5 nm). Three photometers measure the 180-230 nm ultraviolet emission, the 337 nm near-ultraviolet emission, and the 777.4 nm emission with a temporal resolution of 10 $\mu$s.

**Electric field change (dE/dt) measurements.** The VLF/LF sferic data recorded by a multi-station electric field (E-field) sensor array consisting of 13 stations were used to examine the relationship between optical emissions by MMIA and lightning activity[53,54]. Each station was equipped with a vertical E-field antenna (with 3dB-bandwidth of 800 Hz to 300 kHz) and recorded the $dE_Z/dt$ (time derivative of the vertical electric field) for each lightning event. The atmospheric electricity sign convention was used for the E-field sensor. All recordings were synchronized with a GPS clock with sampling rate of 5 MHz. The nearest station (about 100 km) from the NBE-producing thunderstorm of interest is installed in Guangzhou (GZ station, 23.568°N, 113.615°E). The geolocation and source peak current of each lightning event was obtained from the Vaisala LS7000 lightning location networks consisting of 19 sensors[55,56]; the location error of the networks is within 0.5 km and the error of estimated peak current is about 10% by comparing with the results of triggered lightning experiments near Guangzhou[56,57].

**Radar and Cloud top temperature measurements.** The IR brightness temperature at 10.4 $\mu$m of the parent thunderstorm is derived from the latest-generation geostationary meteorological satellite Himawari-8[58]. The target observation provides an image of the thunderstorm every 10 min. The spatial resolution of IR data is 2 km × 2 km. Base reflectivity is derived from a 2.88-GHz S-band Doppler radar with a detection range of about 340 km. The radar was operated with a 6-min cycle producing polar volumes, and the distance between the radar and the thunderstorm is typically about 100 km. The height of tropopause is estimated from the atmospheric sounding data (http://weather.uwyo.edu/upperair/sounding.html).

**Time match.** The time accuracy of ASIM is within ~20 ms due to the time shift. The time provided by ground-based lightning array is assumed to be the correct time of lightning events. To match the event between the two systems, first, we identify all the blue emissions from ASIM during the overflight of the thunderstorm, and the occurrence time of blue emissions can be obtained by subtracting the propagation delay. Then, we search the lightning event from the ground-based lightning array during a time window of about 30 ms. It is noted that the NBEs especially those of negative polarity are isolated with other discharges as seen from the context of sferic waveform, hence, it can exclude optical emissions from other discharges. Table S1 presents the detailed time correction information of NBEs. With this time correction method, we find that nine blue emissions are associated with radio signal classified as negative NBEs and other three blue emissions are associated with positive NBEs. It is noted that the time shift of about 22 ms and 4 ms is similar to previous studies on the ASIM time shift[25,34].

## Data availability

The source data used for Figs. 1–4 in this study are available in the open-access depository (https://doi.org/10.5281/zenodo.5525898). The raw ASIM data are from https://asdc.space.dtu.dk/mmia_observation/. Atmosphere sounding data are available at http://weather.uwyo.edu/upperair/sounding.html. Source data are provided with this paper.

## Code availability

The code that supports the findings of this study is available from the author (feifan@ustc.edu.cn) upon reasonable request.

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

## Acknowledgements

The authors would acknowledge financial support from the National Key Research & Development Program (2017YFC1501501), the Project of Stable Support for Youth Team in Basic Research Field, CAS (YSBR-018), the B-type Strategic Priority Program of the Chinese Academy of Sciences (XDB41000000), National Natural Science Foundation of China (41775004, 41831070, 41875006, 41974181, 42005068, and U1938115), and the Open Research Project of Large Research Infrastructures of CAS—"Study on the inter-action between low/mid-latitude atmosphere and ionosphere based on the Chinese Meridian Project". F.F.L is also supported by Fundamental Research Funds for the Central Universities (WK2080000134). F.F.L. thanks Ms. Jun Shao for her support and encouragement throughout the study. A.L. and D.L. is supported by the European Research Council (ERC) under the European Union H2020 programme/ERC Grant Agreement 681257. F.J.G.V. acknowledges funding by the Spanish Ministry of Science and Innovation (AEI) under project PID2019-109269RB-C43 and the FEDER program. F.J.G.V., A.L., and D.L. acknowledge financial support from the Spanish AEI through the Center of Excellence "Severo Ochoa" award for the Instituto de Astrofísica de Andalucía (SEV-2017-0709). ASIM and the ASIM Science Data Centre are funded by ESA and by national grants of Denmark, Norway and Spain. We thank Vaisala for the GLD360 lightning data.

## Author contributions

F.F.L. processed the ASIM and electric field data, produced the figures, and drafted the manuscript. J.H.L. and B.Y.Z. proposed this work and revised the paper. T.N. led the principal investigator of the ASIM project, and revised the manuscript. G.P.L., N.Ø., F.J.G.V., A.L., V.R. and D.S.L revised the manuscript. O.C. provide the interpretation of the MMIA data. W.T.L. provided the error analysis of Vaisala data. F.F.L., G.P.L., W.T.L. and B.Y.Z. conducted the ground-based electric field measurements. All authors discussed and support the conclusions.

## Competing interests

The authors declare no competing interests.
