## [Peer Review File · Nature Communications]

REVIEWER COMMENTS

Reviewer #1 (Remarks to the Author):

Review of "Optical emissions associated with narrow bipolar events in radio signals from thunderstorm clouds penetrating into the stratosphere" by Liu et al.

This manuscript reports on the simultaneous observation via VLF/VHF and ASIM of positive and negative NBE events in a single thunderstorm. The main conclusion of the paper is a confirmation of earlier studies that both polarities NBE are associated with shorter optical wavelengths, supporting the idea that both phenomena are cold streamer breakdown processes. The data and methods presented in the manuscript are generally sound and convincing. The effect of in-cloud scattering and time-delayed/expanded light emissions is a nice result, as is the simultaneous observation of both polarities NBE. While a nice observation, these results are incremental to our understanding of the physics behind NBE-type breakdown and mostly confirm earlier observations by others.

Suggestions for improvement are below.

Grammar. The manuscript will benefit from a careful re-read by a native English speaking person, to correct use of articles and other words.

Date stamps. Suggest using consistent date formatting, like "7 August 2019".

Time stamps. Suggest writing times as "13:06:20.000" etc (with the additional colon)

Do a find and replace all of "lighting" with "lightning" (but check for those cases where you mean to say "lighting", I think there are none in this manuscript.)

Lines 72-72. This sentence is vaguely worded and does not convey the point of the studies well.

Line 77. array -> arrays, or add 'a'

Line 104 (and one other occurrence). Suggest avoiding putting a reference after a number (8). Perhaps add this reference in relation to the satellite in the methods section only.

Line 107. Add "the" in two places

Line 108. Intense convective surges: I would really like to see an altitude versus time evolution plot of the radar reflectivities through the core of the storm, with the NBE events overplotted, that clearly shows the strong convective surges. This in itself would be a nice result. Can such a figure be added, perhaps in the supplement section?

Line 110. was -> were, if you mean all three positive NBE

Line 136. Correct spelling of "calculable". Or preferably, rephrase as "... and their height also cannot be inferred from the ..."

Line 141. Add 'a' before "range"

Line 145. Correct spelling of "significant"

Line 147. "... suggesting that negative NBEs are associated with fast streamer breakdown." The authors are not wrong, but this conclusion cannot be drawn from this context. All you can say is that the negative NBE are not associated with leaders, but a cold plasma, i.e. streamer breakdown. (Cannot say it was "fast" from these observations.)

Lines 162 and 164. What do you mean by "strength"? Power?

Line 183 and other occurrences of the word "differential". I suggest not using the word differential, and write this as "time-derivative" everywhere instead if that is what is meant.

Line 191. Change "is the one ensuing [...]" to "is the one in which [...] ensue"

Line 230. "below the thundercloud". I think you mean below the top of the thundercloud?

Line 236. Suggest changing "hosting thunderstorms" to "parent thunderstorms".

Line 245. Fix "may be due to that the".

Lines 246-247. Thus, this optical difference is likely due to the shorter channel length. Why would negative NBE at higher altitudes have shorter channel length? Also, I don't follow this conclusion from what is said earlier.

Line 259. Clear linear. I think the results are too sparse to show a clear linear correlation. Perhaps just say "a clear correlation".

Line 271. Remove "the".

Lines 277-281. I think this conclusion is too far-fetched, because negative NBE are not that common (like blue jets). While there may be a link with greenhouse gas concentrations, in my opinion it would be very hard to get meaningful observations that would tell us more about those gases, because there are so many unknowns. I suggest removing this conclusion from the text and also the abstract, since it isn't needed. The NBE observations themselves are impactful enough. If the authors stand by including this conclusion, I suggest expanding on the text and explain how studies relating greenhouse gases and RF detections of cloud-top discharges would to be done.

Line 286. Activities -> activity

Line 290. Define SPS

Line 300. I suggest writing E_z instead of E_v for the vertical electric field component

Line 304. Correct lighting

Line 326. the time window -> a time window

Line 353. Add "data" after "lightning"

Line 354. This website is some kind of portal, in Chinese, and it asks for some code or login, which I do not have. I recommend the authors put the data used for this manuscript in a depository that is in English and does not require a login.

Line 355. Correct spelling of "website"

Line 433. Correct misspelled word "limpact"

Table 1. Suggest sorting all events by increasing time, not decreasing, and not separating out the negative and positive events.

Table 1. Use time stamps hh:mm:ss.ssss instead of hhmm:ss.ssss

Table 1. use correct symbol for greek " μ " in uW

Table 1. The word NULL has no scientific meaning for the VLF height. Say something like "undetermined" or "?" or such, or leave open

Table 1. Perhaps make the title longer, adding "of the NBE observations" or similar wording.

Figure 1a. The radar site should be marked in the top view, because it is very close to being in view of the current plot limits. Suggest making the zoom a little wider, so the radar site can be included and the ISS track is shown longer as well. In addition, add a line in this plan view plot indicating the direction of the radar scan shown in panel b.

Line 532. Correct spelling of "lightning"

Line 537. How is the height of the tropopause inferred? Say it is an estimate, or give a reference or other supporting data.

Figures 2 and 3. I suggest using a color other than light yellow-green for the VLF waveforms. For instance, use black, dark green, medium blue, and red, for the various traces so the traces are better visible on a white background.

Lines 543-544 and 551-552. "is in line with the pink". No idea what is meant by this. I see no pink lines.

Reviewer #2 (Remarks to the Author):

Review of "Optical emissions associated with narrow bipolar events....stratosphere"

This is a good paper which reports new results addressing a long-standing puzzle in lightning science, viz, are NBEs relatively devoid of light emissions (relative to ordinary lightning)? The feat of performing ISS-based multispectral observations of emissions from the tops of thunderclouds is not only impressive but is also tremendously important for the research. Since the negative NBEs take place at the cloud's upper screening layer, only airborne or space-borne optical observations can succeed in viewing the emission region with minimal cloud scattering of light. This would be hard, or likely impossible, to reproduce with ground-based observations. The ground-based observer would see only a temporally-smear-out and attenuated version of the source emission. It would not be easy from the ground to make a compelling association of that highly stretched-out light with the prompt NBE radio signal. By comparison, the ISS-borne observations reported here have an ideal viewing location, which preserves the fast rise time of the blue emissions from the top of the cloud.

That ability to see the emission region without debilitating cloud scattering has also resulted in the lovely result of Figure 4b, a linear relationship between optical output power and estimated NBE current. That would be impossible without a clear view of the emission region, free of cloud scattering.

A few minor quibbles:

Figure 1 caption: "Green dots present the GZ lightning VLF/ELF receiver". Actually not "green dots", but rather a single green triangle.

"The pink cross-present..." should be "The pink X's present..."

Figure 2 caption: "Sferic E-waveform is in line with the pink..." needs some more explanation. Ditto in Figure 3 caption.

Reviewed by:

Abram R. Jacobson (who wishes not to be anonymous)

Reviewer #3 (Remarks to the Author):

The paper "Optical emissions associated with narrow bipolar events in radio signals from thunderstorm clouds penetrating into the stratosphere" provides some useful information on optical signatures of NBEs. The subject under study is new and the data presented is indeed of interest to the lightning community. The paper confirms some previous theories on the optical emissions from NBEs. There are, however, a few issues that need to be addressed, before the manuscript may be acceptable for publication.

Introduction:

Please, mention that NBE is known as Compact intracloud discharges (CIDs) as well. Many papers refer to NBE as Compact intracloud discharges (CIDs) e.g. [1]–[5].

[1] A. Nag and V. A. Rakov, "Compact intracloud lightning discharges: 1. Mechanism of electromagnetic radiation and modeling," *J. Geophys. Res. Atmos.*, vol. 115, no. 20, 2010.

[2] A. Nag, V. A. Rakov, D. Tsalikis, and J. A. Cramer, "On phenomenology of compact intracloud lightning discharges," *J. Geophys. Res.*, vol. 115, no. D14, p. D14115, Jul. 2010.

[3] A. F. R. Leal, V. A. Rakov, and B. R. P. Rocha, "Compact intracloud discharges: New classification of field waveforms and identification by lightning locating systems," *Electr. Power Syst. Res.*, vol. 173, pp. 251–262, Aug. 2019.

[4] T. Wu, W. Dong, Y. Zhang, and T. Wang, "Comparison of positive and negative compact intracloud discharges," *J. Geophys. Res.*, vol. 116, no. D3, p. D03111, Feb. 2011.

[5] F. Lü, B. Zhu, H. Zhou, V. A. Rakov, W. Xu, and Z. Qin, "Observations of compact intracloud lightning discharges in the northernmost region (51°N) of China," *J. Geophys. Res. Atmos.*, vol. 118, no. 10, pp. 4458–4465, May 2013.

It is important to mention what is the sign convention used in the field records (Atmospheric electricity or Physics sign convention) to say that the NBE is Positive or Negative.

Results:

Line 86. This reviewer thinks that the way that is presented the time is confusing. The authors should consider 13:05:56. The same thing happens in other parts of the manuscript.

Line 104. Probably, it is a missing character after "Himawari"

Line 113. Confusing timestamp

Line 137. The first time that the acronym (GZ) is mentioned in the text. Please write up.

Discussions:

Line 229: Would be "shape" instead of "sharpe"?

Methods:

Electric field (E-field) change measurements

It is not clear for this reviewer if the NBE electric field waveforms presented in the paper are indeed E or dE/dt. If the multi-station electric-field sensors record dE/dt, how the integration of dE/dt is performed? Throughout the paper, the authors give NBE waveforms as being E-fields, not dE/dt.

Point to Point Responses to Reviewers' Comments

Reviewer #1:

This manuscript reports on the simultaneous observation via VLF/VHF and ASIM of positive and negative NBE events in a single thunderstorm. The main conclusion of the paper is a confirmation of earlier studies that both polarities NBE are associated with shorter optical wavelengths, supporting the idea that both phenomena are cold streamer breakdown processes. The data and methods presented in the manuscript are generally sound and convincing. The effect of in-cloud scattering and time-delayed/expanded light emissions is a nice result, as is the simultaneous observation of both polarities NBE. While a nice observation, these results are incremental to our understanding of the physics behind NBE-type breakdown and mostly confirm earlier observations by others.

Response: Thanks for your positive comments. We have now thoroughly revised the manuscript based on your suggestions.

Suggestions for improvement are below.

Grammar. The manuscript will benefit from a careful re-read by a native English speaking person, to correct use of articles and other words.

Response: We have corrected the typos and rephrased some sentences to improve the readability.

Date stamps. Suggest using consistent date formatting, like "7 August 2019".

Response: Done.

Time stamps. Suggest writing times as "13:06:20.000" etc (with the additional colon)

Response: Corrected as suggested.

Do a find and replace all of "lighting" with "lightning" (but check for those cases where you mean to say "lighting", I think there are none in this manuscript.)

Response: we replaced all of "lighting" with "lightning".

Lines 72-72. This sentence is vaguely worded and does not convey the point of the studies well.

Response: Thanks for pointing this out. We now revised this sentence as "Positive NBEs inside thunderclouds were found to be associated with the 337 nm emissions rather than the 777.4 nm emissions, suggesting they are corona-like discharges formed by many cold streamers".

Line 77. array -> arrays, or add 'a'

Response: Corrected

Line 104 (and one other occurrence). Suggest avoiding putting a reference after a number (8). Perhaps add this reference in relation to the satellite in the methods section only.

Response: Corrected as suggested

Line 107. Add "the" in two places

Response: Done

Line 108. Intense convective surges: I would really like to see an altitude versus time evolution plot of the radar reflectivities through the core of the storm, with the NBE events overplotted, that clearly shows the strong convective surges. This in itself would be a nice result. Can such a figure be added, perhaps in the supplement section?

Response: This is a good suggestion. The evolution plot of the radar reflectivity through the parent thunderstorm can show the detailed info of the parent thunderstorm. We have added it in the supplement section. As shown in Figure R1, it can be seen that the cloud top develops intermittently above the tropopause penetrating into the stratosphere. Interesting to note that negative NBEs are always produced at the overshooting tops (OT) under the strong convective surges. These cloud top discharges are likely related to the blue emissions (e.g., pixies and gnomes) reported by Lyons et al. (2003) and Chanrion et al. (2017). This good agreement would provide a new electrical tool to detect OT. Of course, the relationship between OT and negative NBEs merits further investigation.

Figure R1 | Evolution of the radar reflectivity through the parent thunderstorm. The blue ‘△’ represents the negative NBEs. The black ‘▽’ presents the positive NBEs.

Line 110. was -> were, if you mean all three positive NBE

Response: Corrected

Line 136. Correct spelling of "calculable". Or preferably, rephrase as "... and their height also cannot be inferred from the ..."

Response: Done

Line 141. Add 'a' before "range"

Response: Done

Line 145. Correct spelling of "significant"

Response: Done

Line 147. "... suggesting that negative NBEs are associated with fast streamer breakdown." The authors are not wrong, but this conclusion cannot be drawn from this context. All you can say is that the negative NBE are not associated with leaders, but a cold plasma, i.e. streamer breakdown. (Cannot say it was "fast" from these observations.)

Response: The reviewer is correct. We deleted "fast".

Lines 162 and 164. What do you mean by "strength"? Power?

Response: Yes, the observed negative NBEs has a wide range of amplitude. Now we revised it as "amplitude".

Line 183 and other occurrences of the word "differential". I suggest not using the word differential, and write this as "time-derivative" everywhere instead if that is what is meant.

Response: We replaced the word "differential" to "time-derivative".

Line 191. Change "is the one ensuing [...]" to "is the one in which [...] ensue"

Response: Done.

Line 230. "below the thundercloud". I think you mean below the top of the thundercloud?

Response: Yes. We revised the description accordingly.

Line 236. Suggest changing "hosting thunderstorms" to "parent thunderstorms".

Response: Done.

Line 245. Fix "may be due to that the".

Response: Corrected.

Lines 246-247. Thus, this optical difference is likely due to the shorter channel length. Why would negative NBE at higher altitudes have shorter channel length? Also, I don't follow this conclusion from what is said earlier.

Response: Thanks for your comments here. The shorter rise time of optical emissions for negative NBEs indicates the relatively short duration. If we assume the breakdown speed does not vary with altitude (Rison et al., 2016), this difference may contribute to the shorter length.

Line 259. Clear linear. I think the results are too sparse to show a clear linear correlation.

Perhaps just say "a clear correlation".

Response: Corrected as suggested.

Line 271. Remove "the".

Response: Done.

Lines 277-281. I think this conclusion is too far-fetched, because negative NBE are not that common (like blue jets). While there may be a link with greenhouse gas concentrations, in my opinion it would be very hard to get meaningful observations that would tell us more about those gases, because there are so many unknowns. I suggest removing this conclusion from the text and also the abstract, since it isn't needed. The NBE observations themselves are impactful enough. If the authors stand by including this conclusion, I suggest expanding on the text and explain how studies relating greenhouse gases and RF detections of cloud-top discharges would be done.

Response: Thanks for your comments. We have removed the related text in the abstract concerning a link with greenhouse gas concentrations under your suggestion. Previous simulation work has evaluated the influence of blue discharges on greenhouse gas in the lower stratosphere. Winkler and Notholt (2015) estimated the production of nitrous oxides to be 10^{12} cm^{-3} of a blue jet at 18 km. The production from a volume of 1 km^3 , with the kilometer-scale blue discharges, would lead to 1660 mole per discharge (Chanrion et al., 2017). This may explain why the concentrations of the various oxides of nitrogen observed in the stratosphere near thunderstorms are higher than the values estimated by models (Park et al., 2004). However, this region of the atmosphere is difficult to access experimentally due to the cloud cover (Lyons, 2006; Chou et al., 2011). Our findings present the direct link between the blue emissions and E-field signal of negative NBEs. This would provide a possible new solution to measure the occurrences and strength of cloud-top discharges with implications for studying the perturbations of greenhouse gas concentrations near the tropopause. The relationship between them merits further investigation in future work.

Line 286. Activities -> activity

Response: Fixed.

Line 290. Define SPS

Response: We add the definition of SPS (second positive system, $C^3\Pi_g \rightarrow B^3\Pi_g$) in the manuscript.

Line 300. I suggest writing E_z instead of E_v for the vertical electric field component

Response: Done.

Line 304. Correct lighting

Response: Done.

Line 326. the time window -> a time window

Response: Done.

Line 353. Add "data" after "lightning"

Response: Done.

Line 354. This website is some kind of portal, in Chinese, and it asks for some code or login, which I do not have. I recommend the authors put the data used for this manuscript in a depository that is in English and does not require a login.

Response: Thanks for the comment. We put the data in the open-access depository (<https://doi.org/10.5281/zenodo.4792277>).

Line 355. Correct spelling of "website"

Response: Done.

Line 433. Correct misspelled word "limpact"

Response: Done.

Table 1. Suggest sorting all events by increasing time, not decreasing, and not separating out the negative and positive events.

Response: We have modified the arrangement of all events by increasing time in Table 1.

Table 1. Use time stamps hh:mm:ss.ssss instead of hhmm:ss.ssss

Response: Changed as suggested.

Table 1. use correct symbol for greek "mu" in uW

Response: Corrected.

Table 1. The word NULL has no scientific meaning for the VLF height. Say something like "undetermined" or "?" or such, or leave open

Response: Changed as suggested. We leave open for the undetermined VLF height.

Table 1. Perhaps make the title longer, adding "of the NBE observations" or similar wording.

Response: Corrected as suggested.

Figure 1a. The radar site should be marked in the top view, because it is very close to being in view of the current plot limits. Suggest making the zoom a little wider, so the radar site can be included and the ISS track is shown longer as well. In addition, add a line in this plan view plot indicating the direction of the radar scan shown in panel

Response: Thanks for your helpful suggestions. Figure 1a has been modified under your suggestions.

Line 532. Correct spelling of "lightning"

Response: Corrected.

Line 537. How is the height of the tropopause inferred? Say it is an estimate, or give a reference or other supporting data.

Response: The height of the tropopause is inferred from the sounding data. We now add it in the supplementary information.

Figures 2 and 3. I suggest using a color other than light yellow-green for the VLF waveforms. For instance, use black, dark green, medium blue, and red, for the various traces so the traces are better visible on a white background.

Response: This is a good suggestion. Figures 2 and 3 have been modified for the better visible.

Lines 543-544 and 551-552. "is in line with the pink". No idea what is meant by this. I see no pink lines.

Response: Thanks for pointing this out. We have revised the description of the caption.

Reviewer #2:

This is a good paper which reports new results addressing a long-standing puzzle in lightning science, viz, are NBEs relatively devoid of light emissions (relative to ordinary lightning)? The feat of performing ISS-based multispectral observations of emissions from the tops of thunderclouds is not only impressive but is also tremendously important for the research. Since the negative NBEs take place at the cloud's upper screening layer, only airborne or space-borne optical observations can succeed in viewing the emission region with minimal cloud scattering of light. This would be hard, or likely impossible, to reproduce with ground-based observations. The ground-based observer would see only a temporally-smearred-out and attenuated version of the source emission. It would not be easy from the ground to make a compelling association of that highly stretched-out light with the prompt NBE radio signal. By comparison, the ISS-borne observations reported here have an ideal viewing location, which preserves the fast rise time of the blue emissions from the top of the cloud.

That ability to see the emission region without debilitating cloud scattering has also resulted in the lovely result of Figure 4b, a linear relationship between optical output power and estimated NBE current. That would be impossible without a clear view of the emission region, free of cloud scattering.

Response: Thanks for your great efforts and positive feedback to our work.

A few minor quibbles:

Figure 1 caption: "Green dots present the GZ lightning VLF/ELF receiver". Actually not "green dots", but rather a single green triangle.

"The pink cross-present..." should be "The pink X's present..."

Response: We have revised the caption of Figure 1 under your suggestion.

Figure 2 caption: "Sferic E-waveform is in line with the pink..." needs some more explanation. Ditto in Figure 3 caption.

Response: Thanks for pointing this out. We have now modified the line color.

Reviewed by:

Abram R. Jacobson (who wishes not to be anonymous)

Reviewer #3:

The paper “Optical emissions associated with narrow bipolar events in radio signals from thunderstorm clouds penetrating into the stratosphere” provides some useful information on optical signatures of NBEs. The subject under study is new and the data presented is indeed of interest to the lightning community. The paper confirms some previous theories on the optical emissions from NBEs. There are, however, a few issues that need to be addressed, before the manuscript may be acceptable for publication.

Response: Many thanks to the reviewer for providing us such constructive suggestions. We have revised the manuscript accordingly.

Introduction:

Please, mention that NBE is known as Compact intracloud discharges (CIDs) as well. Many papers refer to NBE as Compact intracloud discharges (CIDs) e.g. [1]–[5].

Response: Thanks for the information. In the revised manuscript, we included those references in the introduction.

- [1] A. Nag and V. A. Rakov, “Compact intracloud lightning discharges: 1. Mechanism of electromagnetic radiation and modeling,” *J. Geophys. Res. Atmos.*, vol. 115, no. 20, 2010.
- [2] A. Nag, V. A. Rakov, D. Tsalikis, and J. A. Cramer, “On phenomenology of compact intracloud lightning discharges,” *J. Geophys. Res.*, vol. 115, no. D14, p. D14115, Jul. 2010.
- [3] A. F. R. Leal, V. A. Rakov, and B. R. P. Rocha, “Compact intracloud discharges: New classification of field waveforms and identification by lightning locating systems,” *Electr. Power Syst. Res.*, vol. 173, pp. 251–262, Aug. 2019.
- [4] T. Wu, W. Dong, Y. Zhang, and T. Wang, “Comparison of positive and negative compact intracloud discharges,” *J. Geophys. Res.*, vol. 116, no. D3, p. D03111, Feb. 2011.
- [5] F. Lü, B. Zhu, H. Zhou, V. A. Rakov, W. Xu, and Z. Qin, “Observations of compact intracloud lightning discharges in the northernmost region (51°N) of China,” *J. Geophys. Res. Atmos.*, vol. 118, no. 10, pp. 4458–4465, May 2013.

It is important to mention what is the sign convention used in the field records (Atmospheric electricity or Physics sign convention) to say that the NBE is Positive or Negative.

Response: Thanks for the reviewer pointing this out. The Atmospheric electricity sign convention was used for the electric field sensor in our observation; that is, positive NBEs occurring between the upper positive charge layer and main negative charge layer produces the negative initial peak while negative NBEs occurring between the upper positive charge layer and screening charge layer produces the positive initial peak. We note that the waveform in Figures 2d and 3d is reversed as the distance of the y-axis increases in reverse.

Results:

Line 86. This reviewer thinks that the way that is presented the time is confusing. The

authors should consider 13:05:56. The same thing happens in other parts of the manuscript.

Response: The authors have revised the time formatting everywhere in the manuscript.

Line 104. Probably, it is a missing character after “Himawari”

Response: Thanks for this reminder. We double-checked it, and found that it is correct.

Line 113. Confusing timestamp

Response: Thanks. We have modified the time formatting.

Line 137. The first time that the acronym (GZ) is mentioned in the text. Please write up.

Response: Corrected as suggested

Discussions:

Line 229: Would be “shape” instead of “sharpe”?

Response: Yes, it is ‘shape’. Corrected as suggested.

Methods:

Electric field (E-field) change measurements

It is not clear for this reviewer if the NBE electric field waveforms presented in the paper are indeed E or dE/dt . If the multi-station electric-field sensors record dE/dt , how the integration of dE/dt is performed? Throughout the paper, the authors give NBE waveforms as being E-fields, not dE/dt .

Response: Thanks for pointing this out. In the original version, we want to describe the sensor record the electric component rather than magnetic signal. Actually, the multi-station electric-field sensors record electric field derivatives (dE/dt) rather than electric field (E) (Qin et al., 2020). We have revised the description accordingly.

REVIEWER COMMENTS

Reviewer #3 (Remarks to the Author):

Thank you for answer my questions. I am fully satisfied with your response. The subject under study is indeed of interest to the lightning community. The supplementary material and the main manuscript present valuable results. Finally, the paper can be accepted without further revision.

Responses to Reviewer Comments

Reviewer #3:

Thank you for answer my questions. I am fully satisfied with your response. The subject under study is indeed of interest to the lightning community. The supplementary material and the main manuscript present valuable results. Finally, the paper can be accepted without further revision.

Response: We really appreciate the reviewer for their great efforts and positive feedback on our work.